# A Systematic Framework for Collecting Site-Specific Sampling and Survey Data to Support Analyses of Health Impacts from Land-Based Pollution in Low- and Middle-Income Countries

**DOI:** 10.3390/ijerph18094676

**Published:** 2021-04-28

**Authors:** Katherine von Stackelberg, Pamela R.D. Williams, Ernesto Sánchez-Triana

**Affiliations:** 1NEK Associates LTD, Allston, MA 02134, USA; 2E Risk Sciences LLP, Lafayette, CO 80026, USA; pwilliams@erisksciences.com; 3The World Bank Group, Washington, DC 20433, USA; esancheztriana@worldbank.org

**Keywords:** risk assessment, burden of disease, low- and middle-income countries, biomonitoring

## Abstract

The rise of small-scale and localized economic activities in low- and middle-income countries (LMICs) has led to increased exposures to contaminants associated with these processes and the potential for resulting adverse health effects in exposed communities. Risk assessment is the process of building models to predict the probability of adverse outcomes based on concentration-response functions and exposure scenarios for individual contaminants, while epidemiology uses statistical methods to explore associations between potential exposures and observed health outcomes. Neither approach by itself is practical or sufficient for evaluating the magnitude of exposures and health impacts associated with land-based pollution in LMICs. Here we propose a more pragmatic framework for designing representative studies, including uniform sampling guidelines and household surveys, that draws from both methodologies to better support community health impact analyses associated with land-based pollution sources in LMICs. Our primary goal is to explicitly link environmental contamination from land-based pollution associated with specific localized economic activities to community exposures and health outcomes at the household level. The proposed framework was applied to the following three types of industries that are now widespread in many LMICs: artisanal scale gold mining (ASGM), used lead-acid battery recycling (ULAB), and small tanning facilities. For each activity, we develop a generalized conceptual site model (CSM) that describes qualitative linkages from chemical releases or discharges, environmental fate and transport mechanisms, exposure pathways and routes, populations at risk, and health outcomes. This upfront information, which is often overlooked, is essential for delineating the contaminant zone of influence in a community and identifying relevant households for study. We also recommend cost-effective methods for use in LMICs related to environmental sampling, biological monitoring, survey questionnaires, and health outcome measurements at contaminated and unexposed reference sites. Future study designs based on this framework will facilitate consistent, comparable, and standardized community exposure, risk, and health impact assessments for land-based pollution in LMICs. The results of these studies can also support economic burden analyses and risk management decision-making around site cleanup, risk mitigation, and public health education.

## 1. Introduction

Human capital losses attributable to environmental pollution are global, high, and increasing. According to the 2019 Global Burden of Disease study, 11.3 million deaths and approximately 396 million disability adjusted life years (DALYs) were attributed to environmental risks globally [1]. In 2017, the Lancet Commission on Pollution and Health estimated that in the countries with the worst environmental conditions, pollution-related disease was responsible for more than 25% of premature deaths [2]. The actual toll of environmental pollution is likely to be significantly higher. Available estimates on the health effects of pollution are primarily based on a limited number of relatively well-researched pollution categories, including ambient and household air pollution, and lack of access to safe water, sanitation, and hygiene, while chemical pollution is likely to be the most significant and underestimated contributors to the global burden of disease [2,3].

Several studies have documented contaminant exposures and a variety of health outcomes in LMICs, including relating to land-based pollution associated with localized, small-scale industries [4,5,6,7,8,9]. A number of initiatives have helped increase awareness of the importance of addressing chemical contamination. For instance, two of the Sustainable Development Goals (SDGs) are specifically directed at chemicals: target 3.9 to reduce the number of deaths and illnesses from hazardous chemicals and air, water and soil pollution and contamination, and target 12.4 to achieve the environmentally sound management of chemicals and all wastes throughout their lifecycle. 

Reducing the risks of chemical pollution and achieving related SDG targets remains a challenge. Uncertainty about the source of chemical pollution exposures and the relationship to health outcomes results in regulatory impediments. A few studies have researched the public health problems stemming from land-based pollution in LMICS; however, it is difficult to compare their results because of differences in study design, researcher objectives, data collection strategies, contaminants evaluated, sampling and laboratory methods, and target populations [10]. This is important as international donor agencies, such as the World Bank and others, must rely on available exposure and health outcome studies to determine and justify resource allocations for projects aimed at pollution cleanup or control. A more focused and unified approach is therefore needed to understand the disease burden associated with land-based pollution sources in these countries.

Epidemiologic studies and risk assessment approaches have been used in high-income countries to improve scientific understanding of the health effects of chemical pollution exposure. However, conducting full-scale cross-sectional or longitudinal epidemiologic studies with sufficient statistical power to discern quantifiable relationships between exposure and health outcomes is time-consuming and resource-intensive [11,12]. Moreover, these types of studies will not necessarily address critical aspects of exposure drivers and the role of individual behaviors or activity patterns in determining exposures [12]. 

Risk assessment approaches can be used to predict the likelihood of adverse health effects based on measured or modeled exposure concentrations and exposure factors combined with toxicity data [13]. However, these methods lack a direct linkage to measurable health outcomes in communities and typically rely on animal toxicity data and uncertainty or “safety” factors that do not reflect the complexity or level of actual environmental exposures or population susceptibilities, particularly among populations facing multiple co-morbidities [14]. Moreover, while standardized exposure factors (e.g., intake rates, consumption patterns, individual behaviors, lifestyle factors, etc.) are fairly well-described in a few high-income countries, there is a lack of such standardized data from LMICs [15,16].

This document presents a framework that combines the strengths of epidemiologic studies and risk assessment approaches and that can be used to fill critical knowledge gaps that currently hamper responses to chemical pollution. The main purpose of such a framework is to assist in the design of representative studies, including the development of uniform sampling guidelines and household surveys, to better enable linkages among environmental contamination, community exposures, and health outcomes from land-based pollution in LMICs. The framework can be applied to a wide range of small-scale, highly polluting industries, and has already been applied to three that are widespread in LMICs:Small-scale artisanal gold mining (ASGM) predominantly releases mercury (Hg), followed by lead (Pb) and arsenic (As) that are discharged or released to air, soil, and surface water during various phases of the mining process. Hg is transformed to methylmercury (MeHg) in aquatic environments, leading to further pathways of exposure and potential health outcomes.Leather tanning is another complex, resource-intensive process that generates a significant number of by-products, solid waste materials, and large amounts of wastewater. These byproducts can contain chromium (Cr), Pb, As, and cadmium (Cd)—metals that will persist in the environment in their original form and are likely to be found in soil, dust, water, and some agricultural products.Used lead-acid battery recycling (ULAB) consists of dismantling and recycling used batteries, usually acquired from motor vehicles. Contaminants encountered during the battery-recycling process primarily arise from the battery components themselves and include Pb, As, and Cd. These metals can be released into the soil after the batteries are dismantled and discharged as solid waste and wastewater during the separation of components in a water bath. Lead is then smelted and refined, which can release toxic vapor and particulate dust.

The remainder of this paper describes the elements of the proposed framework using the three industry sectors as examples. The document is structured as follows: Section 2 describes the development of customizable conceptual site models (CSMs) tailored to each industry sector (Figure 1). The CSMs allow for a qualitative understanding of site processes and activities, the ways in which communities can be exposed, and the range of potential health outcomes associated with exposures. Section 3 provides a set of key questions and mapping exercises to identify contaminant zones of influence in the community and select representative households and individuals within households for study inclusion using grid and targeted sampling approaches. This is followed by a series of recommendations for collecting sampling and household survey data in a consistent and uniform, yet flexible and practical, manner. Specifically, the document provides guidance on study design, sampling strategies, number of samples, analytical and laboratory methods across environmental and biological media and contaminants, and health outcome assessments. Section 4 presents our conclusions. 

## 2. Conceptual Site Models (CSMs) by Activity

CSMs describe qualitative linkages between contaminant sources and releases and exposures and health outcomes in a population (i.e., source to receptor) by showing how contaminants of concern (CoCs) that are released or discharged from specific activities can migrate through the environment, and the pathways and routes by which individuals in the population can be exposed to those contaminants and experience subsequent health effects [13,14,17]. The CSM captures essential relationships across key components of the analyses and provides a roadmap for quantifying these relationships based on the data collection efforts that are the focus of the systematic framework. 

Characterizing exposures and intake requires information on chemical concentration, frequency and duration of exposure, and various exposure factors such as behavior, time and activity patterns, and contact or intake rates [18,19]. Standardized exposure factors have not been derived for application in LMICs as they have in other countries [15,16,18,20,21,22]. Differing dietary habits, earthen floor housing, climate and time spent outdoors, local environments, dusty conditions during particular times of the year and unpaved roads, and other factors and behaviors all imply that exposure factors, while unknown, are likely to differ from data obtained from high-income countries [23,24].

While environmental monitoring provides data on exposure concentrations, household surveys provide information on intake rates and exposure factors, which, when combined, can be used to quantify potential exposure pathways and routes in a population [13,14]. On the other hand, biomonitoring data provides a measure of internal dose of a contaminant or its metabolite from all sources and exposure pathways [25,26,27]. In the absence of data, mathematical models are used to translate exposure doses to internal doses, including physiologically-based pharmacokinetic modeling (PBPK), as well as different statistical methods ranging from relatively simple regressions and correlations to structural equation modeling [28,29,30]. If both biomonitoring and exposure data are available, then it is possible to apply a variety of statistical and mathematical methods to explore predictors of internal dose [30]. Understanding exposure routes and pathways helps to inform strategies for exposure reduction, and others have called for more intensive studies to link exposures from various pathways to biomonitoring data [25,27,31,32].

Under the proposed framework presented here, developing the CSM is both informed by and leads to (1) an understanding of the site-specific environmental setting to identify the CoC zone of influence in a community; and, (2) the selection of relevant households and individuals within households for subsequent data collection efforts. Quantitatively linking site contamination to population exposures and health effects requires that environmental, biomonitoring, household survey, and health outcomes data be concurrently collected for a subset of households and individuals selected as representative of the population of interest compared to an unexposed reference population. The first step, defining contaminant zones of influence and establishing the boundaries of the study and reference areas, is accomplished by developing land-use maps and collecting information on the local population to better understand the location of people relative to site contamination and activities leading to land-based pollution. The second step, selecting representative households for study and identifying individuals within households upon which all subsequent data-collection efforts will be focused, emerges from the mapping of the environmental setting and delineated zone of influence as well as a preference to target vulnerable or susceptible populations.

### 2.1. Overview of Processes

Figure 2, Figure 3 and Figure 4 provide generalized CSMs for each sector/activity that should be refined in study and site-specific settings. Note that metals are the primary CoCs associated with these informal and small-scale activities and are the focus of health-outcome measurements for participating individuals within households. These general CSMs can be tailored to a specific activity or site by applying a set of checklists (for example, Table 1) together with a mapping exercise to spatially locate activities leading to land-based pollution relative to populations, in particular vulnerable or susceptible populations.

#### 2.1.1. Artisanal Scale Gold Mining (ASGM)

Figure 2 provides the generic CSM for ASGM activities. ASGM is the process of extracting ore from sediments or rock to separate out gold [33]. This process is typically done by hand and includes the following steps:Extraction of ore from alluvial deposits or hard rock via surface excavation and sediment pumping;Separating gold from ore using crushing and milling processes;Concentrating extracted gold using wet or dry methods;Addition of elemental mercury to create mercury-gold alloy (whole ore and concentrate amalgamation);Heating of amalgam to vaporize mercury and separate gold (sponge gold);Additional refining and heating of sponge gold to remove residual mercury and impurities.

Elemental mercury (Hg) and the organic form methylmercury (MeHg) are the primary CoCs at ASGM sites. Depending on the source and composition of the ore being mined, lead (Pb), and to a lesser extent, arsenic (As), may also be present. Vaporized Hg is released during the amalgamation process and emitted into the atmosphere, where the Hg oxidizes and deposits into soil, lakes, rivers and oceans via both wet and dry deposition. This results in the direct inhalation of Hg in ambient air as well as indirect ingestion or dermal contact with Hg from contact with contaminated soil or water. MeHg, transformed from Hg by bacteria in aquatic systems, also readily accumulates in aquatic food webs (e.g., fish and shellfish), leading to exposures in individuals consuming fish and shellfish either recreationally or commercially. As the mined ores are mechanically ground and processed, significant amounts of Pb dust can be released, and this process may occur in residential areas outside the primary mining area as individuals bring chunks of ore home for processing [34,35,36,37] with little differentiation between living and working areas [38]. Dry milling, which is commonly employed during the processing stage, tends to magnify the level of dust produced, and in many areas, processing may occur within housing areas using the same mortars and pestles used to prepare food. Even when this processing occurs outside of residential areas, miners often return home with contaminated clothing. Amalgamation is associated with the highest documented exposures, and in some cases, amalgamation also can occur in a centralized location within residential areas. Additionally, children may travel to the mines to sell food, and will therefore be exposed directly to Pb dust and Hg vapor near the mining sites, and possibly facilitate indirect exposures by bringing unsold (cross-contaminated) food back into residential areas. In addition to airborne transport of Pb dust, the grinding and sluicing process often occurs near water sources, which can result in contamination of surface water with Pb as well as initiating the Hg–MeHg conversion process and subsequent bioaccumulation in aquatic organisms.

Mass balance studies developed by van Straaten [33] in Tanzania and Zimbabwe showed that most Hg losses occur during the amalgamation phase, releasing between 70–80% of Hg to the atmosphere and between 20 and 30% lost to tailings, soils and water. This study also found that Hg concentrations in soil tend to be highest within several km of amalgamation activities, although it is not clear this is a generalizable phenomenon.

#### 2.1.2. Used Lead Acid Battery Recycling (ULAB)

Figure 3 provides the generic CSM for ULAB activities. ULAB recycling consists primarily of dismantling and recycling used batteries, usually acquired from motor vehicles [39,40]. The general steps in the process include:Collection and transportation of used batteries to a recycling area;Dismantling of used batteries, which can be done by children and not necessarily localized to a central location;Separation of component battery parts in a water bath so that Pb sinks to the bottom and plastics rise to the top;Air drying of Pb and Pb oxide-containing materials; mixing material with coal, soda ash, and scrap metal; and transferring material to uncovered vessels for heating;Smelting and refining of lead components in open kettles, with only rudimentary pollution control equipment or protective gear;Washing and shredding or melting of plastic components. These plastic components are then repurposed in the community;Purification and treatment of sulfuric acid electrolyte (fluid from the batteries);Treatment and disposal of waste products (e.g., plastics, residual lead, water).

Contaminants and other materials encountered during the ULAB recycling process arise primarily from the battery components themselves, although some additional materials are added during the smelting and refining process. Specifically, Pb, As, and Cd are components of batteries as are lead oxide, battery acid, and plastics. Water is typically added to separate battery components and coal, soda ash, and scrap metal are added during smelting and refining [39,40].

Pb, As, and Cd are released to onsite soils as batteries are dismantled. Pb vapor and particulate (including fly ash) are released to air and soil throughout the smelting process as the kettles are filled and emptied. Solid waste containing Pb, As, and Cd is generally discharged to unlined lagoons or pits, where leaching and erosion can occur, while wastewater containing Pb, As, and Cd is discharged to onsite soils or even nearby surface waters.

#### 2.1.3. Small Scale Leather Processing and Tanning (Tanneries)

Figure 4 provides a generic CSM for tanning activities. Informal and small-scale tanning and leather processing operations are a complex, resource-intensive process known to generate a significant number of by-products, including a variety of contaminants such as metals [9,41,42]. Although the specific tanning process is unique to each facility and will influence the exact contaminants that are generated, the key steps include:Delivery of raw hides and skins;Sorting and trimming of raw hides and skins;Curing and storage of skins;Soaking of skins involving detergents, enzymes, water, and biocides;Unhairing and liming using various deliming agents, surfactants, enzymes, and solvents;Pickling and tanning involving chromium (Cr) and aldehydes;Retanning and dyeing with dyes, Cr, and vegetable tans;Drying and mechanical finishing;Coating, which can involve the use of synthetic coating materials and solvents.

Contaminants originate from across the inputs used at various stages of the process, including deliming agents, enzymes, surfactants, solvents, chromium, salts and acids, aldehydes, and dyes. The primary CoC at tanning facilities is Chromium (Cr). Cr is found in different states in the environment, predominantly Chromium III (CrIII, an essential nutrient) and under certain environmental conditions Chromium VI (CrVI, an unstable form of Cr that is not an essential nutrient and highly toxic). The tanning process relies on chromium salts primarily in the form of chromium sulfate, and although the salts are not particularly hazardous in and of themselves, the process and added constituents (e.g., lime, salts, acids, fungicides) facilitate the oxidation of CrIII to CrVI under certain environmental conditions. Other primary CoCs at these sites include As, Pb, and Cd. Other constituents, such as organic solvents, biocides, and finishing agents may also be present but are not the focus of the current example.

Contaminants originating from tannery sites, particularly metals, may enter the environment during various phases of the tanning process. Frequently, there is onsite storage of solid waste residues and sludge containing organic materials including protein, fat, hair, dirt and process chemicals. At many locations, there will be offsite use of these organic solid waste residues and sludge as livestock feed or sludge fertilizer for agricultural crops [43,44]. Protein, fat, and waste organic skin materials are sometimes ground, dried and used as concentrated protein feed, particularly for chickens, with significant amounts of CoC residues. In some cases, solid wastes may be incinerated, leading to CoC releases to air from the incineration process, as well as ash disposal. Tanning activities generate large amounts of wastewater containing salts, process chemicals, and organic solid wastes, releasing Cr, Pb, As, and Cd to soil or surface water, as well as impacting chemical oxygen demand (COD) and biological oxygen demand (BOD). Migration of CoCs can impact local agriculture or drinking water [42,45,46,47].

### 2.2. Identify Contaminant Zone of Influence (Characterize Environmental Setting)

For each site under study, a land-use mapping exercise should be undertaken to define the relevant geographic area (e.g., village, town, city) that locates all physical source areas relative to other infrastructure or places where populations, particularly vulnerable or susceptible populations such as children, spent the most time (e.g., residential areas, housing units, schools, town center, community areas). This task will help define the potential zone of influence or footprint associated with site activities by delineating the spatial boundaries that connect the site-specific CSM within the local community. For example, information on local hydrology and ground water, the location of aquifers and ground water flow, disposition and potential migration of waste water from the point of discharge to ultimate exposures, use of waste water or contaminated surface water for irrigation or livestock use, or direct contamination of drinking water sources is compiled based on guiding questions as shown in Table 1. A similar exercise should be undertaken for the unexposed reference population to ensure that local pollution sources (resulting in the release of the same CoCs) are not present.

### 2.3. Select Participating Households and Individuals

In addition to understanding the environmental setting in which localized activities occur, information is needed on local population characteristics to identify the specific households and individuals within households for further study. This includes collecting information on population size and density (e.g., village, urban, peri-urban), age/gender distribution, wage and labor statistics, and noting locations where sensitive subpopulations spend time within the study area. This information is used to establish the fraction of the local population participating in local activities, and to identify the distance of residential and other areas from processing areas, noting that in some cases, these kinds of small-scale activities occur within residential or community areas. The final list of sampling locations will depend on where individuals within participating households spend their time, particularly within and around the home and public areas such as schools, community centers, places of worship, and playgrounds. A similar exercise should be undertaken for the unexposed reference population.

The framework recommends a combination of grid and targeted-sampling to identify participating households and individuals within households. While randomized, grid-based sampling represents the gold standard [48,49], it is important to reconcile the need for sampling within vulnerable or susceptible populations that may not be captured using a grid-based method alone. For example, exposures to Pb, Hg, MeHg, and As are associated with neurodevelopmental outcomes in children. Therefore, children represent an important population of concern across all of the sectors. Similarly, given the importance of preconception and perinatal exposures, women of child-bearing age also represent a particular population of concern. 

Determining the exact grid size for selecting households (which will impact the sample size) will require some flexibility, depending on the site-specific CSM, population density, environmental characterization, and resource constraints [50,51,52,53,54]. For example, at a site with known wastewater discharges to a stream that flows several kilometers downstream from the source area and ultimately discharges to a pond, it may be beneficial to select both households within 1 km of the source area as well as other households from downstream locations. As another example, sites that are densely populated within a smaller geographic area may require a smaller grid size than sites with households that are spread out over a larger geographic area. CoCs originating from these kinds of activities have the potential for fairly far-reaching impacts depending on how far away on-site wastes are being repurposed (tanneries, for example), as well as the details of wastewater discharges, and ultimately will depend on the environmental setting and the site-specific attributes of the exposed population. Satellite-based approaches may be helpful, and publicly-available algorithms specifically designed for LMICs are available [49,55].

Depending on the scenario, recommended grid densities range from 20 × 20 m to 100 × 100 m, with most falling generally in the 40 × 40 m to 60 × 60 m range over 1–2 km. A household is selected from each grid node, and if a selected household is not willing to participate in the study, a neighboring household in the same grid space should be chosen. Sample sizes for environmental sampling (e.g., soil, dust, agricultural products, water, and fish/sediment) depend on the selected grid size [56,57,58]. With respect to identifying individuals within households for study inclusion, the framework recommends the following preferred hierarchy: (1) children under age 10, (2) pregnant women or women of child-bearing age, followed by (3) other adults of any age and gender living in the household.

The framework focuses on several predominant metals generated by these sectors and emphasizes exploratory associations across multiple endpoints or objectives (e.g., environmental contamination, individual exposures, several possible health outcomes per CoC). Given the varying objectives and disparate data sources, it is challenging to conduct a single statistical power calculation to determine optimal sample sizes. Therefore, the framework recommends conducting site-specific power calculations (e.g., [56,59]) based on the study-specific hypotheses and objectives in the context of available resources and anticipated analyses. In general, power calculations involving contaminant exposures and health outcomes will depend on the statistical approach(es) to be used in analyzing the data, anticipated effect sizes, and/or the difference between two populations [56,57,59,60], including, (1) anticipated probability of a health outcome given no exposure (general prevalence in the population); (2) anticipated relative risk; (3) confidence level; (4) significance level; and, (5) relative precision. For studies with other objectives, e.g., building local capacity or focused on a single domain such as biomonitoring, power calculations should be more straightforward given less complex objectives and data sources. As a reasonable compromise, the framework recommends that for a typical study, the number of households selected should range between 100 and 400 (average of 200 to 300 households) per site, based on the example calculations provided in the framework documents.

Several simplified examples including unique attributes for each site-specific context are presented in Appendix A, along with a checklist of questions to identify localized activities specific to the process in question as they relate to potential population exposures. 

## 3. Data Collection and Sampling Recommendations

Once the sampling locations, households, and individuals within households have been identified, the framework provides specific recommendations for data collection and sampling methods with an emphasis on in-field and non-invasive (e.g., requiring blood draws or other invasive approaches) sampling strategies. Recommendations are provided for the design and implementation of environmental sampling, household surveys, biomonitoring, and health outcome measurements, using a combination of self-reported, laboratory, clinical, and survey-based methods. These recommendations are based on a review of the literature to establish the state-of-the-science across each of these domains, and then applying a set of criteria to identify the most pragmatic approach for use in LMIC settings. A similar exercise should be undertaken for the unexposed reference population.

### 3.1. Environmental Sampling

Under the proposed framework, environmental samples provide the data to link exposure concentrations in the environment with biomonitoring and health outcome data from individuals across participating households. Many studies from the literature rely on limited environmental sampling and assume that, for example, a small number of non-representative samples collected at a particular location (usually closest to the pollution source) provides an appropriate measure of exposure applicable to all individuals in a community [10]. Analyses based on limited data of this kind will not provide a reliable basis for linking to specific health outcomes and biomonitoring-based exposure concentrations. Thus, the framework emphasizes collecting exposure data that will both provide an overview of environmental concentrations as experienced in the community, as well as specific exposure concentrations as experienced by study participants. Some of the recommendations are generalizable across industries, while others are specific to the predominant CoCs associated with that sector as described below and summarized in Table 2. Ideally, comparable environmental sampling data should be collected from an unexposed reference population, recognizing that limited resources may only allow for a direct comparison based on biomonitoring data alone.

Dermal contact and incidental, direct, and indirect ingestion of contaminated surface soils are the primary routes and pathways of exposure to CoCs from the three processes evaluated here. These are experienced by both adults and children, although the latter are more likely to have direct and more frequent contact with surface soil because of their behaviors and activity patterns, and there is growing evidence that children in LMICs are more highly exposed via this pathway [15,61]. Dermal exposures occur when adults or children walk barefoot on surface soil or their bodies touch this soil (e.g., during play outdoors). Incidental ingestion can occur when individuals get soil on their skin (e.g., fingers) or an object (e.g., toy), which then comes into contact with their mouth or food. Direct ingestion can occur when individuals eat dirt or soil (this is a common practice among some children and generally still involves the top layer of soil), whereas indirect ingestion can occur when crops are grown in contaminated soil (e.g., below-ground root vegetables) or are impacted by fugitive dust or airborne soils (e.g., above-ground leafy vegetables), discussed in Section 3.1.4

#### 3.1.1. Soil Samples

Soil sampling locations should be based on where the participating individuals spend the most time, as guided by the sampling approach mentioned above and information obtained from the administered household survey (see next section). For example, if a participating child routinely plays in the backyard and/or community playground instead of the home or community garden, this is where soil samples should be collected (or vice versa for participating women, if applicable). Properly calibrated portable XRF (PXRF) analyzers are now routinely used for in-field measurement of metals, particularly in resource-constrained settings [62]. Thus, the framework recommends use of an in-field PXRF analyzer to analyze individual samples, followed by confirmatory sampling in a laboratory setting for a subset of composited samples. Although PXRF provide in-field data, there are some limitations, including higher detection levels as compared to laboratory detection levels [63,64]. PXRF analysis of Cr at tanning sites, in particular, is challenging, with detection levels 5–10 times higher than laboratory detection levels and an inability to distinguish between total Cr, CrIII, and CrVI [65].

Regardless of process or activity, the framework generally recommends collecting at least 4 individual samples from each sampling location and evaluating these samples using an in-field XRF for, at a minimum, the primary metals (e.g., Pb, As, Cd, Hg, and Cr, depending on specific aspects of the process) and electronically recording those results. Following the in-field evaluation, the samples should be composited and sent to an accredited laboratory (local, if possible, or split samples to assist in building local capacity) to undergo a multimetal screen. Further recommendations specific to the predominant CoCs associated with each process are noted in Table 2. Tanning activities, for example, are associated with numerous contaminants, but the most significant identifiable CoC associated with health outcomes is CrVI. Because this speciation requires an expensive and specialized analysis, the recommendation is to analyze a smaller subset of samples for CrVI to account for resource constraints in LMICs.

#### 3.1.2. Dust Samples

Small-scale activities in LMICs, such as those evaluated here, can result in direct releases of vapors and particulate that settle on the ground as well indirect releases of fugitive dust from contaminated surface soil. Dermal contact, incidental ingestion, and inhalation of contaminated dust can all occur, and indoor exposures to contaminated dust (tracked in from outdoors) is of particular concern due to the duration, frequency, and proximity of contact with indoor surfaces. In some cases, indoor dust and soil may be indistinguishable (e.g., soil-based flooring), and sampling of these surfaces may be combined.

In general, the framework recommends collecting two dust samples from indoor surfaces such as floors, tables, and windowsills. Dust sampling locations should be based on where the participating individuals spend the most time or are most likely to come into contact with contaminated house dust, as determined by the household survey. For example, if a participating child sleeps or routinely plays on the bedroom floor, this is where a dust sample should be collected. Specific methods for sampling dust will differ depending on the surface substrate. For dwellings with dirt floors, methods analogous to soil sampling should be used. For dwellings with impervious and smooth surfaces (e.g., wood floors, wood tables, windowsills), wipe samples are preferred (e.g., GhostWipe™). In some instances, vacuum sampling may be required, such as for rough (e.g., brick, stone, etc.) or carpeted surfaces. As with soil, it is recommended that individual samples first be analyzed in-field using PXRF, and then composited and sent to an accredited laboratory for a multimetal screen.

#### 3.1.3. Water Samples

Local water supplies may be impacted from the small-scale activities considered here due to leaching or runoff of contaminated soils to surface water or groundwater, leaching of waste products from lagoons or pits to surface water or groundwater, or migration of contaminated surface water from wastewater discharges to other surface water sources or groundwater. Dermal contact and ingestion of contaminated water are the primary exposure routes and pathways, although contaminated water may also be used for irrigation, leading to potential exposures via consumption of agricultural products [44,66,67,68]. Either surface water or groundwater (or both) can be used as sources for drinking water and it is not uncommon to see site-related wastewater discharges directly to surface water sources used in different ways in the community [66,69]. Specific aspects of each of the identified processes may also contribute to increased exposures, for example, the tanning process involves many different inputs that significantly alter the pH of receiving waters and facilitate the conversion of CrIII to CrVI, the toxic form of Cr. Similarly, while the ASGM process involves the use of elemental Hg, it is the microbial conversion of Hg to MeHg in aquatic environments that leads to bioaccumulation, potential fish contamination, and resulting exposures to humans via fish consumption. These are all identified through the CSM development process.

Water samples should be collected at the individual household or dwelling where the water is available for consumption (e.g., drinking) or use (e.g., bathing) if there is a municipal water supply. In many cases, drinking water sources will be obtained from a communal location such as a community well that serves multiple participating households. Although ideally water samples would be collected from the location where the water is stored following transport (e.g., container inside home or school), this will be resource-intensive and it will suffice to collect samples from the communal location to leverage data across the largest number of participating households. Specific to ASGM sites, water samples should also be collected from the water body or bodies where fish are caught that are consumed by participating households.

In general, PXRF is not suitable for aqueous sampling as it measures the metal concentration in the particulate matter and thus requires preconcentration with solid-phase extraction disks [70,71,72]. A recent study found that the use of waterproof films at a maximum 2–4 mm depth effectively utilized PXRF, but the method is only suitable for “high” concentrations of metals [73]. Thus, the framework recommends collecting 1-liter samples, to be sent to an accredited laboratory for a multimetal screen rather than utilizing in-field sampling, recognizing it may be possible to use PXRF for in-field water sampling as the technology matures [74].

#### 3.1.4. Agricultural Product Sampling

In communities affected by the processes and activities under evaluation, contaminated irrigation water from impacted surface water or ground water sources may be used on crops for human or animal consumption, or crops may be grown in contaminated soil or subject to aerial deposition of CoCs, with resulting uptake in root systems or deposition on foliage [75,76]. Additionally, animals may be grazed on contaminated soil or given contaminated water to drink [76,77,78]. Examples of locally produced agricultural products in LMICs include fruits, vegetables, and grains; animals consumed for meat (e.g., chickens, cattle), and various animal products (e.g., milk, cheese, eggs). These products may be produced at multiple scales, ranging from small family gardens at the household level to large commercial operations that sell products at local or off-site markets [66,69]. Consequently, direct ingestion of contaminated agricultural products may represent an important exposure route and pathway.

Sampling of agricultural products should focus on what is most commonly consumed by the study participants, which will be informed by the survey-based data collection (e.g., food frequency questionnaire). Examples of different types of foodstuffs that might be sampled at individual sites include:Rice grown in surface water impacted by wastewater from tanning, mining, or smelting activities;Chickens foraging directly at processing sites, particularly for tanning activities;Root vegetables grown in soils from backyard gardens irrigated with surface water impacted by wastewater;Leafy greens grown downwind within a depositional area;Beans or other legumes grown in soils (e.g., fertilized with organic solid wastes from tanning operations or irrigated with contaminated surface water);The framework recommends that agricultural product samples should be sent to an accredited laboratory for a multimetal screen. The exact biomass required for sampling will need to be determined by the laboratory or laboratories involved in the analyses. Although it is possible to use PXRF for agricultural products, the method does require dried and powdered samples and may not provide an appropriate level of precision [79,80].

#### 3.1.5. Fish/Shellfish and Sediment Samples

Fish/shellfish (and potentially sediment) sampling is only recommended for ASGM sites, and not explicitly recommended for tanning and ULAB sites as the primary CoCs associated with these industries are not known to biomagnify in the same way as MeHg, and fish consumption is unlikely to represent a primary exposure pathway and route for these sites. 

Fish/shellfish samples should target those species consumed by participating individuals or households. However, unlike with agricultural products, rather than sampling fish/shellfish obtained from each household, fish should be collected from the water body in which they are caught to provide the range of concentrations that all fish/shellfish consumers are likely to experience and to decrease the overall number of required samples. Targeted sampling of the potentially exposed population (e.g., recreational anglers) may be used to identify specific households to include in the sampling program. Alternatively, if there is a commercial fishery, fish may be obtained from local anglers. 

The framework recommends collecting synoptic samples of fish/shellfish, water, and, if resources allow or bioaccumulation modeling will be conducted, sediment from the same water body. Fish species should be those that are ecologically important in the aquatic food web (e.g., forage fish that serve as prey base for larger fish), which is particularly important if bioaccumulation modeling will be conducted. Depending on resource availability and the specific objectives of the study, the framework recommends collecting and analyzing 10 individual fish/shellfish (species consumed by participating households) or composite samples of a minimum of three individual fish/shellfish for MeHg. If bioaccumulation modeling will be conducted, additional samples should be collected representing differing trophic levels (e.g., forage fish, bottom-feeding fish) specifically to support model development. In these cases, collecting synoptic sediment samples will also be required. 

### 3.2. Household Surveys

The framework recommends administering a standardized household survey to all participating individuals in the exposed and unexposed (reference) populations (if possible) to assist in guiding the field sampling efforts, to obtain exposure factor data, and demographic and socioeconomic data [81,82,83]. The survey questionnaire can also be used to obtain information on some aspects of health outcomes, particularly with respect to self-reported symptoms, formal medical diagnoses, and other potential confounders (e.g., nutritional status). Exposure factor data includes intake rates and time-activity patterns that can be combined with exposure concentrations from the environmental sampling to estimate population exposures or predict biomonitoring data. The household survey will also provide key information on occupational and residential history and other factors relevant for subsequent analyses, particularly to stratify results by demographic variables and to address confounding factors. 

The standardized household survey includes questions about time spent at home versus other locations (e.g., school, public areas), time spent within the home (e.g., sleeping), household characteristics (e.g., flooring material), drinking water intake (L/day), sources of water, and other lifestyle and time and activity behaviors contributing to potential exposures. It also includes a food frequency questionnaire (e.g., 24-hour diary together with an estimate of the previous month) to both categorize the diet, and to determine the most appropriate local agricultural (and/or fish) products to sample that will leverage exposures across the highest number of households. Note that it is imperative that the survey questions and general manner in which the questions are asked and available response categories remain relatively consistent across studies to facilitate comparisons across studies in the future.

The framework provides an example of a standardized questionnaire including categories of questions (e.g., time-activity, economic, demographic) as well as specific questions within categories. The design relied on recommendations from Aday and Cornelius [81], which provides details for designing health surveys using high-quality, effective, and efficient statistical and methodological practices, optimal sample designs as well as guidance on statistical theory for survey data analysis, and is well-recognized as a standard reference for survey design. The household survey is the primary instrument for collecting data on potential confounding variables, including demographics, smoking, lifestyle, and occupational history that may influence exposures to the CoCs.

### 3.3. Biomonitoring

Under the proposed framework, biological samples are used to quantify total exposure to the CoCs from all sources and exposure pathways and routes for each participating individual. The greater use of this approach in evaluating human exposures in research and regulatory contexts has been advocated [26,32,84,85,86]. Biological sampling data can also be used to confirm or validate estimates of CoC exposure based on the exposure factor data from the household surveys combined with environmental sampling data associated with each participating household. Additionally, biological samples can be used to analyze for possible indicators of health outcomes or nutritional and health status. That is, the goal of the biological sampling design is to quantify the magnitude of exposure among individual population members to each CoC (e.g., biomarkers of exposure), but depending on the sampling matrix, there may also be the opportunity to identify potential subclinical evidence of disease that has a high probability of being associated with exposures to CoCs (e.g., biomarkers of effect). To the extent possible, the biomonitoring data will be used to link environmental contamination from specific industries in LMICs to community exposures and health outcomes.

The framework recommends collecting biological samples from all study participants, if possible, including the unexposed reference group. Biological sampling matrices include urine, blood, toenails, and hair, and these are rated according to a hierarchy as described below. While breast milk and cord blood represent additional matrices, these are not recommended due to their limited utility in relating exposure to health outcomes for an infant population. Each of the recommended matrices offer advantages and limitations depending on the contaminant, health outcome or intermediate health outcome, and biomarker to be measured. An emphasis is placed on point-of-care (e.g., comparable to in-field for environmental sampling) methods that provide rapid, immunoassay-based results [25,27,87,88].

The key factors that were considered when evaluating preferred biomarkers include: (1) how well the biomarker correlates with the dose (or external exposure) to appropriate forms of the contaminant (e.g., total Cr vs CrIII vs CrVI; MeHg vs. Hg); (2) how well the biomarker correlates with the contaminant concentration in tissue relative to the health outcome; (3) how well the biomarker measurement correlates with changes in the effective dose at the target tissue over time; (4) an understanding of the cultural characteristics of the population; (5) technology availability; and (6) invasiveness of sample collection.

Accordingly, the framework categorizes and ranks different biomarkers as follows:“Gold standard”—This biomarker has been well-vetted in the literature with one or more validated, cost-effective laboratory methods with high levels of precision. This is the preferred biomarker given the primary research objectives in this document.“Screening level”—This biomarker is an appropriate default for low-resource applications. It is the least invasive, lowest cost, typically using in-field XRF. However, only the total metal can be measured with high detection levels, and may not have the precision to evaluate statistical associations with outcomes.“Low preference”—This biomarker can be used as a last resort, but is generally not preferred due to limitations with respect to associations (i.e., they are not the best measure of exposure and/or predictive of outcomes based on literature studies).“To be avoided”—This biomarker is not recommended as it doesn’t measure the exposure of interest, is expensive, and/or doesn’t have a validated method.

While PXRF measurements of toenails represent a reasonable measure of exposure for most metals, there are limitations to this technology, particularly with respect to quantitatively linking exposures to health outcomes. This is due to both high detection levels as well as an inability to speciate metals, introducing uncertainty and the potential for error when analyzing relationships to health outcomes. 

Table 3 provides an overview of preferred biomonitoring matrices based on these categories as well as health outcome measurement for each CoC.

### 3.4. Measuring Health Outcomes

Given the goal to explore relationships between health outcomes and associated environmental contamination and exposures from land-based pollution in LMICs, health outcomes are organized by the CoC with which they are associated, recognizing that several of the CoCs have the potential for overlapping health outcomes and combined risks (e.g., exposures to MeHg, Hg, Pb, and As are all associated with neurodevelopmental outcomes in children) (see Table 3). If possible, the same types of health outcomes should be measured for comparison purposes in the unexposed reference population.

Conduct age-specific, culturally-relevant cognitive testing for each child

Besides collecting self-reported symptoms as part of the household survey, the framework addresses three types of possible health outcomes that can be measured. The first is a medical diagnosis related to direct or measurable clinical outcomes known to be associated with exposure to the CoC of interest (e.g., bladder cancer or hyperkeratosis associated with As exposures; cognitive deficits as measured by age-specific standardized testing instruments associated with exposures to Pb and As). The second is an intermediate, non-specific observation or measurement associated with the health outcome of interest (e.g., increased blood pressure associated with cardiovascular outcomes that may be related to exposure to Pb and Cd). The third is an intermediate, specific biochemical measurement associated with the health outcome of interest (i.e., biomarker of effect) that requires laboratory or in-field analysis of a biological matrix (e.g., diagnosis of anemia based on hematocrit level in blood that may be related to Pb exposures; micronucleus formation in blood that may be associated with genotoxic effects of As). Note that the various testing instruments, particularly those designed to assess children’s development or cognitive function, must be validated in the context for which the instrument is being used. This is one area in which the recommendation to “standardize” test methods will require flexibility and site-specific modifications.

Although it would be desirable to measure unique health outcomes associated with exposures to each CoC from one or more of these categories, a key challenge of this type of investigation is that the primary CoCs from the activities under evaluation (e.g., Cr/CrVI, Pb, and As) as well as other factors share common biological targets, so it is difficult to discern the relative contribution (if any) of each CoC exposure to the identified health outcome. For example, exposure to multiple CoCs has been associated with cognitive and neurodevelopmental outcomes in children using age-specific standardized instruments (e.g., Bayley’s Scale of Infant Development, IQ tests). Additionally, intermediate measures of health outcomes in the absence of overt toxicity (biomarkers of effect) may show associations with exposure concentrations as measured through biomonitoring and/or environmental concentrations of CoCs. Therefore, it is important to note that while biomarkers of exposure are CoC-specific, biomarkers of effect may not be CoC-specific. Moreover, there are many other factors that could influence the same health outcomes associated with these CoCs, ranging from lifestyle factors such as diet, exercise, and smoking status to common environmental exposures such as air pollution. It is anticipated that these latter factors will be captured during the household survey and subsequently controlled for through statistical analyses of the data and comparisons to the unexposed reference population.

## 4. Discussion 

Here we present a uniform, yet customizable, framework for study design and data collection and sampling related to ASGM, ULAB and tanning activities in LMICs. While the example CSMs presented here are specific to these activities, the framework and process is generalizable to any activity of interest (e.g., e-waste) and provides a structured approach for data collection and sampling starting with problem formulation to establish the conceptual and spatial boundaries of the analysis (i.e., zone of contaminant influence). This approach is consistent with the movement towards “fit for purpose” risk evaluations that emphasize the upfront problem formulation step in the environmental health sciences [60,89,90,91,92].

The framework also centers on identifying participating households and selecting individuals within households using a combination of the site-specific CSM and grid and targeted sampling approaches in the context of localized mapping. These are standard approaches used in environmental settings to identify relevant CoCs, exposure routes and pathways, and sampling locations [17,18,19]. These exercises set the stage for detailed recommendations for CoC-specific sampling and analysis methodologies across individuals within participating households including collecting environmental samples across a range of environmental media, administering detailed household surveys, collecting biomonitoring data using preferred matrices, and measuring health outcomes based on a combination of self-reported symptoms and diagnoses, biomarkers of effect, and clinical evaluations. Standardizing these approaches, to the extent possible, provides a consistent basis for evaluating potential community- and population-level exposures and impacts across different geographic areas and sites. The framework can also be modified to inform additional objectives, such as building local capacity or focused analyses that address only one domain (e.g., population biomonitoring without environmental sampling, or just environmental sampling of specific media). That is, the framework is designed to standardize data collection and analyses, yet provide flexibility in implementation to achieve alternative objectives and better inform decision-making at the local to global level in LMICs.

The framework explicitly does not provide recommendations on the choice of statistical models and methods for analyzing the data that will be collected because such decisions will depend on site-specific study objectives and CSM, as well as the number of individuals/households and samples available. Instead, the framework is directed toward achieving the primary objective of collecting a uniform set of data across these different domains (e.g., environmental exposure, human behavior, biomonitoring, and health outcomes), which will subsequently be combined and evaluated to explore the burden of disease associated with a specific sources of environmental contamination (e.g., ULAB, ASGM or tanning activities). Within each domain and across domains, many different statistical and modeling approaches are available to explore possible correlations and associations depending on the specific quality and quantity of data collected, study objectives, and availability of analytical tools. These may include different kinds of regression models, odds ratios or relative risk calculations, logistic models, statistical vs mechanistic and process models, and Bayesian approaches. There is a greater likelihood of reliably combining data across domains, and the results of studies conducted at different times and places with varying objectives, if the studies follow consistent study design and data collection methods, as has been noted by a large international consortium under the auspices of the World Health Organization [93] (see also http://gather-statement.org/ (accessed on 27 April 2021)). Consequently, the framework is designed to ensure optimal data collection and sampling to better inform research goals and decision-making more broadly rather than provide prescriptive recommendations around analyses.

The framework document recommends collecting analogous data from a reference or unexposed population in order to discern differences attributable to particular exposure sources (e.g., the particular industrial activity such as ULAB, ASGM, or tanning). The biomonitoring data, which provides a measure of internal exposure regardless of exposure source, may be most informative in this regard. If monitored levels are comparable across populations, that may suggest ubiquitous diffuse sources of contamination.

It is important to recognize that health outcomes in LMICs are influenced by many factors, including the quality of health care services that are offered, as well as social determinants of health, such as access to nutritious food, access to water and sanitation, decreased levels of sanitation and shelter generally, infectious agents, and lack of appropriate medical care [94,95]. These factors, along with co-morbidities, can interact with contaminant exposures from land-based pollution and may, all things equal, ultimately play the larger role in determining population health status [92,94,96]. Despite these challenges, acquiring a better understanding of the health impacts associated with land-based contamination attributable to particular processes and activities requires collecting data specific to those objectives, and standardizing these collection efforts will facilitate combining resulting data across locations or with other kinds of data (e.g., economic costs). In addition, it would be useful to develop a mechanism for compiling the collected data and making it accessible to researchers and decision-makers for future use and analysis, another recommendation emerging from the World Health Organization sponsored effort on Guidelines for Accurate and Transparent Health Estimates Reporting (GATHER; http://gather-statement.org/ accessed on 27 April 2021). For example, time-activity and behavioral data obtained through the household survey should be compiled into country- or region-specific databases to support development of risk assessments and other analyses that require quantitative exposure factors (e.g., body weight, food consumption) to predict contaminant-specific intake rates applicable to analyses for pollution sources beyond those considered here. 

The state of the science continues to evolve and is moving at a faster pace than regulatory guidelines developed to evaluate public health risks [13]. For example, biomonitoring is a rapidly expanding field with improvements in molecular techniques leading to the identification of novel biomarkers, including oncogenes, tumor suppressor genes, microRNAs and long non-coding RNAs, DNA methylation and others [97,98,99,100,101]. The evolving discipline of “omics,” including proteomics and genomics, has also led to the identification of genetic and epigenetic alterations, typically based on blood samples and utilizing various laboratory-based assays, that may be associated with health outcomes, particularly those occurring later in life [97,98,99,100,101,102,103,104]. Additionally, the concept of the Exposome has emerged as a complement to the genome, which strives to measure individuals’ total exposure (internal and external) over a lifetime and understand how such exposures and gene-environment interactions ultimately impact public health [86,105,106,107,108]. Although these novel approaches are not yet mature enough to recommend for routine use in evaluating exposures and health risks in LMICs, their progress and development should be closely monitored and considered in the future. A key drawback at the current time is the requirement for specialized laboratory equipment, invasiveness of biological sampling (typically a venous blood sample is required), and the increased expense of such analyses.

Finally, a number of decisions around implementation of the recommendations emerging from the framework will necessarily remain in the purview of the field-based team, which requires flexibility. Notably, building local capacity to carry out data collection is an important consideration, and in some contexts, may be the primary objective. This is not necessarily an all-or-nothing proposition, consequently, the framework endeavors to provide opportunities for developing a local vested interest in data collection. For example, sending composited environmental samples to an accredited laboratory following in-field measurements allows for both immediate data collection but also confirmation and verification of the field results. Building local capacity effectively requires participation from one or more in-country laboratories.

## 5. Conclusions

The framework proposed here seeks to fill a critical gap that hampers responses to growing chemical pollution challenges across LMICs. It provides a standardized approach for linking environmental contamination from land-based pollution associated with specific small-scale industrial activities in LMICs to community exposures and health outcomes at the household level. Operationalizing the framework will require significant capacity building, particularly because a number of decisions around implementation of the recommendations emerging from the framework will remain in the purview of the field-based team. Given that chemical pollution affects millions of people across the world, most of them low income and vulnerable groups, standardizing data collection programs will allow for stronger collaboration and integration across disciplines to better support decision-making around land-based pollution sources and the relationship between these sources of contaminant exposures and health outcomes at the community level. As data are collected and analyzed, and ideally made publicly available in a consistent, anonymized format, it should be possible to identify a set of reduced variables that could be used on an ongoing basis to evaluate land-based pollution sources and health outcomes at smaller scales. 

(The framework as applied to three sectors are available as separate published reports from the World Bank website)

## Figures and Tables

**Figure 1 ijerph-18-04676-f001:**
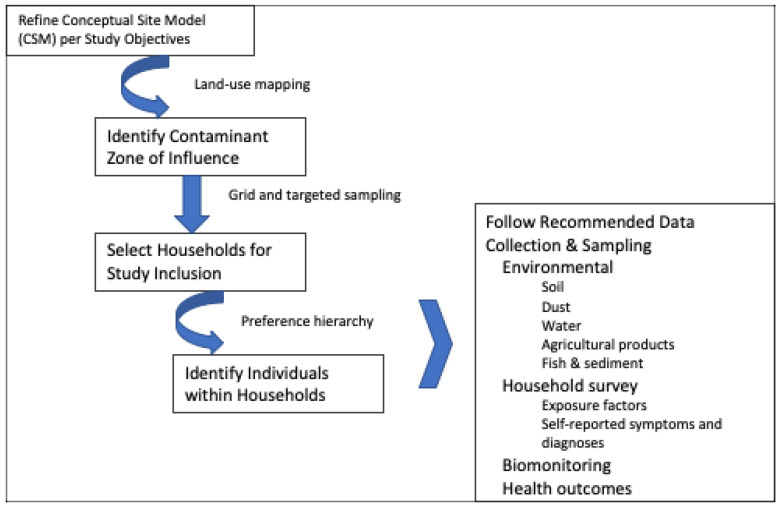
Overview of framework to support health impact studies in LMICs.

**Figure 2 ijerph-18-04676-f002:**
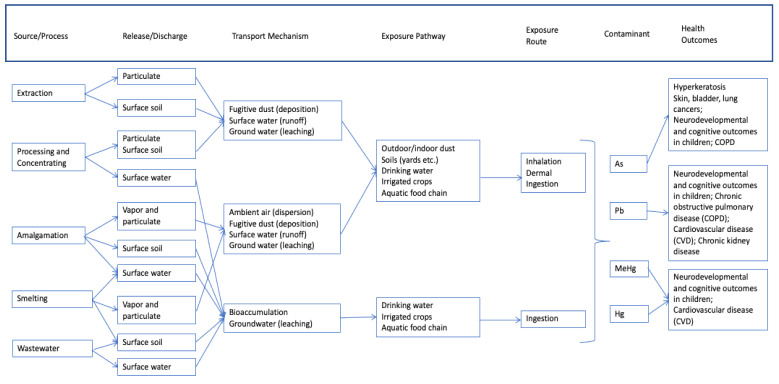
Generalized conceptual site model for artisanal scale gold mining activities.

**Figure 3 ijerph-18-04676-f003:**
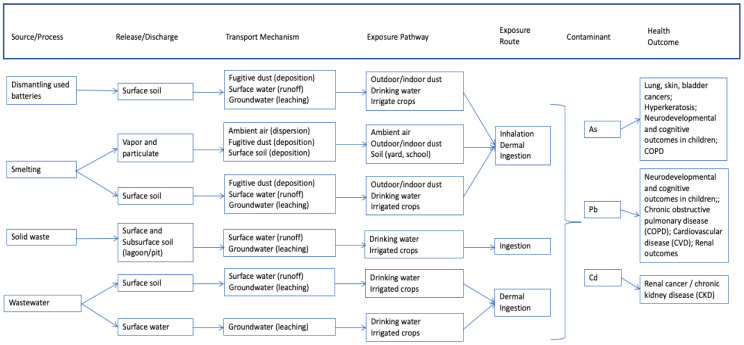
Generalized conceptual site model for used lead acid battery recycling activities.

**Figure 4 ijerph-18-04676-f004:**
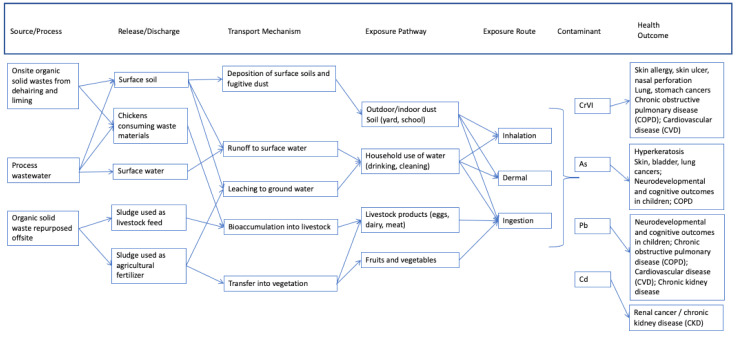
Generalized conceptual site model for small-scale leather processing and tanning activities.

**Table 1 ijerph-18-04676-t001:** Overview of sector-specific guiding questions.

ASGM	ULAB	Tanning
Locate ASGM activities in the context of local populations, noting where different aspects of the process may occur. In some areas, grinding and milling occurs in local homes. Identify locations of all surface waters, including ditches, creeks, streams, rivers, and lakes. Identify what is known about ground water, depth to the water table and aquifers in the study area. Identify the prevailing wind direction, particularly relative to residential areas, local waterbodies, and small- or large-scale agricultural activities within several km of primary site activities, particularly amalgamation. Identify water bodies within a depositional area of ASGM activities, or impacted by wastewaters or soil runoff. Identify agricultural areas, community gardens, and the potential for backyard gardening. Locate sources of irrigation water that might be impacted by ASGM discharges, including direct or indirect surface water discharges or releases to soils that can runoff or erode. Establish whether ground water is used for irrigation and whether there is a leaching pathway. Identify locations where animals or animal products (e.g., milk, eggs) are raised for consumption.	Locate ULAB activities in the context of local populations, noting where different aspects of the process may occur. In some areas, battery breaking may occur in separate areas from primary smelting and refining. Identify locations of all surface waters, including ditches, creeks, streams, rivers, and lakes. Identify what is known about ground water, depth to the water table and aquifers in the study area. Identify the prevailing wind direction, particularly relative to residential areas, local waterbodies, and small- or large-scale agricultural activities. Dispersion and deposition of lead dust and other metals is likely to be significant, and can occur over large areas. Identify water bodies within a depositional area of ULAB activities, or impacted by wastewaters or soil runoff, both of which are likely to contain lead and other metals. Identify agricultural areas, community gardens, and the potential for backyard gardening. Locate sources of irrigation water that might be impacted by ULAB wastewater discharges, including direct or indirect surface water discharges or releases to soils that can runoff or erode. Establish whether ground water is used for irrigation and whether there is a leaching pathway. Identify locations where animals or animal products (e.g., milk, eggs) are raised for consumption.	Locate small-scale tanning activities in the context of local populations. Note whether process activities are dispersed in different areas, for example, curing and soaking occurring in one location while fleshing and liming occurring elsewhere. In some cases, specific activities will be clustered within smaller neighborhoods. Identify locations of all surface waters, including ditches, creeks, streams, rivers, and lakes. Identify what is known about ground water, depth to the water table and aquifers in the study area. Identify the prevailing wind direction, particularly relative to residential areas, local waterbodies, and small- or large-scale agricultural activities. Identify agricultural areas, community gardens, and the potential for backyard gardening. Locate sources of irrigation water that might be impacted by tanning discharges, including direct or indirect surface water discharges or releases to soils that can runoff or erode. Establish whether ground water is used for irrigation and whether there is leaching pathway. Identify locations where animals or animal products (e.g., milk, eggs) are raised for consumption. Organic wastes from tanning, including residual scrap hides, protein, hair and fur, dung, fatty material, and other organic solid wastes, including chemicals from the tanning process, are often repurposed as either livestock feed or fertilizer. Identify and locate these activities on a map.

**Table 2 ijerph-18-04676-t002:** Overview of sector-specific sampling guideline recommendations.

Media	ASGM	Tannery	ULAB
Soil	If Pb-based ores are used and resources allow, 25% of randomly selected household samples and 50% of targeted samples undergo bioavailability testing for Pb	25% of randomly selected household samples and 50% of targeted samples undergo laboratory analysis for CrVI If resources allow, 25% of randomly selected household samples and 50% of targeted samples undergo bioavailability testing for As, Pb	If resources allow, 50% of randomly selected household samples and 100% of targeted samples undergo bioavailability testing for Pb
Dust	No additional specific recommendations	25% of randomly selected household samples and 50% of targeted samples undergo laboratory analysis for CrVI	No additional specific recommendations
Water	25% of randomly selected samples and 100% of community drinking water samples undergo laboratory analysis for MeHg	25% of randomly selected household samples and 100% of community drinking water samples undergo laboratory analysis for CrVI	No additional specific recommendations
Agricultural Products	Contaminated water used as irrigation water and/or airborne or soil deposition are the most common pathways by which agricultural products can become contaminated	In addition to contaminated water used as irrigation water and deposition, tanning activities lead to large amounts of organic wastes, which may be used as fertilizer with little additional processing	Contaminated water used as irrigation water and/or airborne or soil deposition are the most common pathways by which agricultural products can become contaminated
Fish & Sediment	See text	Not required for tanning sites	Not required for ULAB sites

**Table 3 ijerph-18-04676-t003:** Overview of activity-specific sampling guideline recommendations.

CoC	Biomonitoring (Exposure)	Health Outcomes
As	Gold standard is metabolite monomethylarsonic acid (%MMA) obtained from a speciated creatinine-adjusted urine sample	Conduct age-specific, culturally-relevant cognitive testing for each child Conduct in-field screening for keratosis on the soles of the feet as part of the household survey or as part of a more formal medical examination If keratosis is observed, consider a carcinogenic biomarker such as DNA adduct assay or micronucleus formation assay Measure C-reactive protein as a non-specific biomarker of intermediate effects on the renal and cardiovascular systems
Cd	International consensus on use of creatine-adjusted urine	Measure sensitive urinary biomarkers, including β2-m (urinary β2-microglobulin), and glomerular filtration rate (GfR) If elevated, consider measuring additional carcinogenic biomarkers, such as DNA adduct formation or micronucleus formation
Cr (CrVI)	Red blood cells or urine; can speciate. Recommend hair given non-occupational exposures	Evaluate CrVI-induced ulceration of the nasal septum mucosa and potential for skin allergies as demonstrated through skin rashes on the hands and feet If dermatological symptoms are observed, consider patch-testing for individuals with dermatological symptoms Consider measuring carcinoembryonic antigen (CEA; a non-specific biomarker of gastrointestinal cancers) in individuals with dermatological symptoms Conduct limited pulmonary function testing (PFT) and measure C-reactive protein (CRP; a non-specific inflammatory biomarker in blood associated with lung, kidney, and cardiovascular outcomes)
Hg	In-field PXRF toenails	Administer the Chronic Inorganic Mercury Intoxication checklist (Doering et al. 2016) to each participant (can be done in-field with appropriately trained personnel or as part of a more formal clinical assessment) Measure protein urea (e.g., albumin)
MeHg	Hair	Conduct age-specific, culturally-relevant cognitive testing for each child
Pb	Venous blood is the gold standard; dried capillary blood spot also used, allows in-field LeadCare Analyzer	Measure blood pressure in adults in the field or as part of a medical examination Measure specific biomarkers including proteinuria (e.g., albumin), anemia status (e.g., hematocrit), cardiovascular risk (e.g., C-reactive protein) Conduct age-specific, culturally-relevant cognitive testing for each child

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
