# Peer review of "A Systematic Framework for Collecting Site-Specific Sampling and Survey Data to Support Analyses of Health Impacts from Land-Based Pollution in Low- and Middle-Income Countries"

_ijerph, 2021, doi:10.3390/ijerph18094676_

Round 1

Reviewer 1 Report

Authors provide a very well-written, comprehensive framework with which environmental health researchers can design risk assessment studies in LMICs. This framework will undoubtedly be of use to research scientists at the beginning stages of their study design process. I commend authors for the organized and very easy-to-follow manner they present this structure. I only have a few, relatively minor suggestions for authors to consider. 

The citations included in lines 388-389: "Sample sizes for environmental sampling (e.g., soil, dust, agricultural products, water, and fish/sediment) depend on the selected grid size [56,57]," don't seem to correspond with grid-based environmental sampling. If that's the case, these citations can be omitted here. 

The suggestion made in lines 409-411

As a reasonable compromise, the framework recommends that for a typical study, the number of households selected should range between 100 and 400 (average of 410
200 to 300 households) per site.

seems somewhat arbitrary. No justification is provided for these numbers. Authors should consider elaborating on why this would be a "reasonable compromise." 

Along these lines, in Table 2, the recommendations for the proportion of samples to include in lab or bioavailability testing also seems arbitrary and unsubstantiated. It would be helpful if readers were made aware of the justification for these numbers (either by including relevant citations in-text or statistical justification in the supplement to the article). 

Although cursorily mentioned in the Discussion section (lines 789-793), the framework failed to describe the identification of potential confounding variables as one of the components of the study design process. Perhaps this should be briefly discussed, for instance, in an additional section denoted "3.5 Identifying Potential Confounding Variables."

Reviewer 2 Report

This is a very well written and well presented manuscript. The proposed framework for health impacts assessments for land use applications is thoughtful and logical, and will provide helpful guidelines for use in research and practice. Minor revision suggestions to clarify and improve the manuscript are provided as follows:

Line 419: Define/explain what is considered as non-invasive sampling.

Tables 1 and 2: Define abbreviations, such as ASGM, ULAB.

Line 650: Consider adding to discussion possible use of saliva for biological matrix sampling. 

Reviewer 3 Report

Dear Authors

Hope you are well during this pandemic period.

About your manuscript I find very interesting your study. In the section introduction maybe you could end with a phrase and not with a figure. Also in the conclusion section you could improve it because is to short regarding the other sections. Nevertheless your study is very well designed, very clear and objective, I consider it an excellent contribution.

Stay safe, best regards
